# Co-Use of Alcohol and Cannabis: Longitudinal Associations with Mental Health Outcomes in Young Adulthood

**DOI:** 10.3390/ijerph18073652

**Published:** 2021-03-31

**Authors:** Kara Thompson, Maria Holley, Clea Sturgess, Bonnie Leadbeater

**Affiliations:** 1Department of Psychology, St. Francis Xavier University, 103 Annex, 2323 Notre Dame Ave., Antigonish, NS B2G 2W5, Canada; x2015jfi@stfx.ca; 2Department of Psychology, University of Victoria, Greater Victoria, BC V8W 2Y2, Canada; cleastur@uvic.ca (C.S.); bleadbea@uvic.ca (B.L.)

**Keywords:** SAM, alcohol, marijuana, mental health, concurrent use, simultaneous use, young adults

## Abstract

Increases in cannabis use among young people has heightened concern about the potential interactive health effects of cannabis with other drugs. We examined the longitudinal association between concurrent and simultaneous (SAM) co-use of alcohol and cannabis in young adulthood on mental health symptoms, substance use behaviors, and substance-related harms two years later. Data were drawn from Time 5 (T5; *n* = 464; 46% male) and 6 (T6; *n* = 478; 45% male) of the Victoria Healthy Youth Survey. At T5, 42% of participants used alcohol-only, 13% used concurrently, 41% used SAM, 1% were cannabis only users, and 3% abstained from cannabis and alcohol. Boys were more likely to use SAM. Higher T5 SAM use frequency was associated with heavier use of substances, more substance-related harms, and symptoms of psychosis and externalizing problems at T6. T5 Concurrent use was associated with conduct symptoms, illicit drug use, and alcohol use disorders at T6 relative to alcohol-only use. Cannabis is commonly used with alcohol and the findings suggest that any co-use (concurrent or simultaneous) may be problematic in young adulthood. Public health messages need to explicitly inform consumers about the possible consequences of using both alcohol and marijuana and the addictive pharmacological impact of using them together.

## 1. Introduction

Alcohol and marijuana use are highest during young adulthood (ages 18–24) than at any other point in the lifespan; 83% of Canadian and 81% of U.S. young adults use alcohol and 33% [1] and 37% use cannabis. Research remains largely focused on the independent associations of alcohol and marijuana use with health and social outcomes (e.g., risky driving, mental health problems) (e.g., [2,3]. However, these substances are commonly used together [1,4], either concurrently (CAM: use of both substances but at different occasions) or simultaneously (SAM: use of alcohol and marijuana on a single occasion). Approximately a quarter of U.S. high school students, and a third of Canadian students report past-year SAM use [5]. SAM use tends to be more prevalent than concurrent use [6] and is of particular concern because of emerging evidence that the simultaneous co-use of alcohol and marijuana results in pharmacological interactions that alter the pharmacokinetic effects of both drugs and results in greater impairment than is experienced with either substance alone [7]. Canada legalized the sale of recreational marijuana nationally in October 2018, and currently 10 US states have legalized retail use [8]. As a result, understanding the consequences of co-use of alcohol and marijuana on health outcomes for young people has become increasing important and would inform the development of targeted educational messages and cannabis policy.

Research on the consequences of SAM use at the between-person and within-person level suggest that SAM is associated with higher and more frequent levels of alcohol and marijuana use and with numerous alcohol-related consequences, including unsafe driving behaviors [9,10,11], social problems, substance use disorders, and cognitive problems compared to non-users or to youth who exclusively use either alcohol or marijuana, even after controlling for sociodemographic factors [5,12,13,14,15,16,17]. However, results from recent within-person studies suggest the observed associations between SAM use and negative consequences may largely be due to increases in alcohol consumed on SAM days, rather than SAM itself [16,17,18].

Studies directly comparing outcomes between SAM and CAM and the relation between SAM use frequency and outcomes are more limited. Several studies have found that both SAM and CAM users experience comparable alcohol and marijuana related risks [17,19]. Others studies have found that SAM use (versus CAM use) is associated with heavier alcohol and cannabis use [1,5] and some (but not all) negative alcohol-related consequences. For example, in a recent study of US college students aged 18–24, SAM use was associated with an overall greater number of negative alcohol consequences than CAM, however risk for each individual specific consequence did not differ between SAM and CAM users after use and covariates were accounted for, except for blackouts [14]. Moreover, Subbaraman & Kerr [6] found that SAM use was associated with more risky driving behaviors compared to CAM use, but not more social harms. Similarly, Linden-Carmichael et al., [15] found that more frequent SAM use among young adults was associated with heavier substance use and negative consequences relative to less frequent SAM use. However, Jackson et al. [14], found no differences in outcomes based on frequency of SAM use. It remains uncertain if SAM use is associated with additional risk for these outcomes beyond CAM and whether there is a dose–response relationship between SAM use frequency and negative consequences.

Another current gap in our knowledge is the associations between SAM use and mental health outcomes. We know that alcohol and cannabis use disorders are highly comorbid with other mental health conditions [20]. Further, heavy use of alcohol and chronic marijuana use patterns are independently associated with a variety of internalizing and externalizing symptoms for young people [21,22]. A study with the current sample showed that youth who initiated marijuana early and used more than once a week across adolescence and adulthood reported higher levels of anxiety, depression and conduct problems ten years later [2] (Thompson et al., 2018). Moreover, there is a consistent link between marijuana use and risk of psychosis across the lifespan [23,24]. There is good reason to suspect that SAM use may pose unique and elevated risks for mental health beyond the concurrent or independent use of alcohol or marijuana alone given the expected more intensive pharmacological effects of combining substances [7,12]. Yet, whether SAM use is associated with symptoms of internalizing and externalizing problems for young people remains unclear.

In the current study we contribute to the research on patterns of co-use of alcohol and marijuana in young adulthood using data from Time 5 (2011) and Time 6 (2013) of a community-based sample of young adults who were followed across a decade (2003–2013). We extend previous research in three ways. First, we examine the previously unexplored associations between SAM use and symptoms of young adult mental health problems and substance-use outcomes two years later compared to alcohol-only users and CAM users. There were too few marijuana only users in our sample to make comparisons to using marijuana alone. Second, we examine the association between frequency of SAM use (i.e., how often substances are used simultaneously rather than just any SAM use) and mental health and substance use outcomes. Third, we examine whether gender moderated the associations between SAM use and mental health and substance use outcomes. Gender differences in the risks associated with co-use of alcohol and marijuana have not been established. While studies consistently report that males are more likely to engage in SAM use than females [1,6,15,16,19], no studies to date have examined the moderating effect of gender on SAM use and associated risks. We hypothesized that SAM use would pose greater risk for mental health symptoms and substance use outcomes relative to alcohol-only use and CAM use and that higher SAM frequency would further accentuate these risks. Give the lack of research on gender differences, no a priori hypotheses were made regarding the moderation.

## 2. Materials and Methods

### 2.1. Participants and Procedures

The Victoria Healthy Youth Survey (V-HYS) was a 10-year longitudinal panel study of youth who were ages 12 to 18 in 2003 (T1; *n* = 662; 48% male; M = 15.5 years old, SD = 1.9). Youth were recruited from a random community sample of 9500 households in Victoria, British Columbia, Canada. There were 1036 households identified with a youth aged 12 to 18 years (M. 15.52, SD. 1.93). Of eligible households, 185 (18%) parents or guardians refused permission, 187 (18%) adolescents refused to participate, and 2 (0.0001%) participants were older than 18 years at the time of interview, leaving a Time 1 (T1) sample of 662 adolescents (342 female). The sample was 85% White, 4% Asian, 4% mixed/biracial, 3% Aboriginal, and 4% other (e.g., Black, Hispanic, or other). Youth were followed biennially over 10 years (i.e., for six assessments; T6; *n* = 478; 45% male; M = 25.8, SD = 2.0).

Written consent for participation was given by youth and the parent or guardian for youth under age 18. Participants received a gift certificate at each interview. A trained interviewer administered the V-HYS individually in the youth’s home or another private place. To enhance privacy, a portion of the V-HYS questionnaire dealing with drug and alcohol use was self-administered and placed in a sealed envelope not accessible to the interviewer. The research protocol was approved by the university’s research ethics board. Attrition was assessed by testing for differences in T1 study variables between youth who remained in the study at T6 (*n* = 478) and those who did not participate at Time 6 (*n* = 184). Lost participants were more likely to be male, χ^^2^(662) = 8.77, *p* = 0.003, have lower socioeconomic status (SES), F (1, 636) = 19.39, *p* = 0.001, be smokers, F (1, 660) = 3.82, *p* = 0.05 and have higher T1 ADHD symptoms (F (1, 660) = 5.63, *p* = 0.02).

### 2.2. Measures

#### 2.2.1. Mental Health Symptoms

The Brief Child and Family Phone Interview [25] was used to assess depressive symptoms (e.g., Feel hopeless?), anxiety symptoms (e.g., Worry about doing the wrong thing?), ADHD symptoms (e.g., Jump from one activity to another?), oppositional defiant disorder symptoms (ODD)(e.g., Argue a lot with others?), and Conduct problems (CD)(e.g., damaged public or private property that did not belong to you). Each disorder has six items and has demonstrated strong psychometric properties with the present sample (see [26]. Items are rated on a 3-point Likert scale (0 = never, 1 = sometimes, and 2 = often) and summed (ranges = 0–12). For conduct problems responses were collapsed to reflect 0 = never and 1 = once or twice due to the low occurrence of ratings exceeding 1. Psychotic symptoms were assessed using the Symptoms Checklist 90-Revised [27]. Ten items were scored yes (0) or no (1) and summed (range 0–10).

#### 2.2.2. Substance Use

Substance use variables included cigarette use (number of cigarettes consumed in the past week), and last 12 months frequency of alcohol use, heavy episodic drinking (HED; frequency of drinking more than five standard drinks), illicit drug use (frequency of using six illicit drugs: hallucinogens, amphetamines, club drugs (ecstasy, K, GHB), inhalants, cocaine, and heroin), and marijuana frequency. Responses to cigarette use were coded 0 (none) to 4 (a full pack or more in the past week). HED, alcohol frequency, marijuana frequency, and illicit drug use were coded 0 (never) to 4 (more than once per week). To assess simultaneous use participants were asked “How frequently did you use alcohol and marijuana at the same time (within 3 h of each other) in the last 6 months”. As an indicator of any simultaneous use, the responses were recoded into ‘yes’ (1 or more instances of SAM use) and ‘no’ (no SAM use). Frequency of SAM use was recoded into “never” (= 0), “1–9 times” (= 1), “10–19 times” (= 2), “20–29 times” (= 3), and “30 or more times” (= 4).

Using measures of alcohol frequency, marijuana frequency and simultaneous use, participants were coded into five groups: abstainers (no use of alcohol or marijuana), alcohol-only (alcohol use but no marijuana use), marijuana-only (marijuana use but no alcohol), simultaneous users (co-use of alcohol and marijuana together within the span of three hours), and concurrent users (both alcohol and marijuana use, but no simultaneous use).

#### 2.2.3. Substance-Related Harm

Social harms were assessed using six items from the Harmful Effects of Alcohol Scale adapted from the Personalized Alcohol Use Feedback Scale (http://notes.camh.net/efeed.nsf/feedback, accessed on 17 November 2020) included: “In the last 12 months, was there ever a time that you felt your alcohol use had a harmful effect on your (1) friendships and social life; (2) physical health; (3) outlook on life; (4) home life or marriage; (5) work, studies or employment; and (6) financial opportunities.” Response options were 0 (no) and 1 (yes). Items were summed and then dichotomized into no harm (0) and at least one harm (1). Cannabis use disorder (CUD) and Alcohol use disorder (AUD) symptoms were assessed using the 10 item Mini International Neuropsychiatric Interview [28]. Responses were coded as 0 (no) or 1 (yes) and summed for marijuana and alcohol use separately. DSM-5 criteria for a substance use disorder, responding YES to two or more items, were used to indicate a substance use disorder. Driving/riding drunk and Driving/riding high was assessed by asking participants four items that assessed how many times during the past 30 days they drove a vehicle when they had been drinking or using marijuana or rode in a vehicle with someone who had been drinking or using marijuana. For each type of impaired driving/riding, responses were combined and coded as 0 = no/never or 1 = yes/ever.

### 2.3. Analysis Plan

Inferential statistics (ANOVA and chi-square) were used to assess group differences on variables of interest. Multivariate regression was used to investigate the association between substance use patterns at T5 (*n* = 464; ages 20–26; response rate = 70%) and substance use and mental health outcomes at T6 (*n* = 478; ages 22–29; response rate = 72%), controlling for baseline (T1) measures. Models were run using Mplus version 8.0 [29] (1998–2017) and separate models were run for each set of dependent variables (mental health outcomes, substance use, and harms). Using dummy coding, we compared concurrent use and SAM use patterns to alcohol-only use (reference group) in Model 1. We compared concurrent use and SAM use in Model 2. In a third set of models, we examined how frequency of SAM at T5 was associated with T6 outcomes. Concurrent and alcohol-only users were coded as zero. All models controlled for gender, baseline age, parent SES (coded using the Hollingshead Occupational Status Scale [30], and baseline measures of alcohol and cannabis frequency, dependent variables where available covariances between dependent variables were estimated. Standard indices were used to assess model fit (i.e., Root Mean Square Error of Approximation (RMSEA) = 0.05 and Comparative Fit Index (CFI) = 0.95) [31]. To account for missing data, models were fit using a Full-Information Maximum Likelihood (FIML) which allows individuals to contribute any information they have available. Robust Maximum Likelihood estimator (MLR) was used to address any non-normality. For each model, gender was examined as a moderator using multiple-group models comparing changes in model fit resulting from imposing and releasing equality constraints on model parameters across groups [32].

## 3. Results

### 3.1. Demographics of Co-Use Groups

At T5, 42% of the sample used only alcohol in the last 12 months, 1% used only marijuana, 13% used both alcohol and marijuana but not at the same time (concurrent use), 41% reported simultaneously using alcohol and marijuana (SAM use), and 3% of the sample were abstainers. Table 1 shows demographics and T6 means of the dependent variables for each substance use pattern. SAM use was higher among boys (59%) than girls (41%). Age of onset for alcohol and marijuana use was approximately one year earlier for concurrent use and SAM use compared to alcohol-only use. SAM use had an average age of onset of co-use of 16.3 (SD = 2.43) and reported simultaneous use of alcohol and marijuana on average 15 times in the last 6 months (SD = 23.40). On average, alcohol-only use was associated with the lowest levels of alcohol and marijuana use during adolescence (T1) and the fewest symptoms of externalizing problems and psychosis, the lowest levels of substance use, and fewest substance-related harms at T6. Generally, SAM use was associated with the highest mean levels of psychosis, substance use and related harms.

### 3.2. Mental Health Outcomes

Table 2, Model 1 reports associations of concurrent use and SAM use with mental health symptoms at T6 compared to alcohol-only use. Compared to alcohol-only use, concurrent use was associated with higher levels of CD and SAM use was associated with higher levels of psychosis, ODD and CD. There were no gender differences (∆χ^2 = 72.63, ∆df = 58, *p* = 0.09). Table 2, Model 2 compared concurrent and SAM use. Compared to concurrent use, simultaneous use was associated with less anxiety and depression (marginally significant). Gender-specific analysis revealed this was only true for boys (anxiety: ß = −0.23, *p* = 0.01; depression: ß = −0.22, *p* = 0.04), not girls. For girls, there were no differences in mental health outcomes between concurrent use and SAM use.

### 3.3. Substance Use

Table 2, Model 1 examined differences in T6 substance use outcomes for concurrent and SAM use compared to alcohol-only use. Compared to alcohol-only use, concurrent use was associated with more illicit drug use and SAM use was associated with more heavy drinking, tobacco, and illicit drug use. Gender differences were significant, ∆χ^2^ = 107.23, ∆df = 24, *p* < 0.001. For boys, substance use did not differ between concurrent and alcohol-only use, but male engaging in SAM use reported more heavy drinking, tobacco, and illicit drug use compared to alcohol-only use. For girls, both concurrent use and SAM use was associated with more heavy drinking and illicit drug use compared to alcohol-only; SAM use was also associated with more tobacco use (∆χ^2^= 107.23, ∆df = 24, *p* < 0.001).

Model 2 compared T6 substance use behaviors between SAM use and concurrent use. Compared to concurrent use, simultaneous use was associated with more heavy drinking, marijuana use, and illicit drug use (*p* = 0.07). Gender differences were significant (∆χ^2^ = 70.84, ∆df = 30, *p* < 0.001) and showed boys engaging in SAM use reported more heavy drinking, marijuana use and illicit drug use compared to concurrent use; female SAM use was associated with more marijuana than concurrent use.

### 3.4. Substance-Related Harm

Table 2, Model 1 compared substance-related harms at T6 across use groups. Compared to alcohol-only use, concurrent use was associated with greater odds of an AUD and SAM use was associated with greater odds of social harms, risky driving related to alcohol and an AUD. There were no differences between concurrent use and SAM use. However, a chi-square test showed that SAM use was associated with cannabis use disorder compared to concurrent use (χ^2^= 15.96, df = 1, *p* < 0.001); Only 5 concurrent users met the criteria for a CUD. There were no gender differences (Model 1: Wald test = 5.266, df = 6, *p* = 0.51; Model 2: Wald test = 6.76, df = 4, *p* = 0.15).

### 3.5. Associations with Frequency of Simultaneous Use

Table 3 reports the associations between frequency of SAM use and each set of dependent variables. For mental health symptoms, higher frequency of SAM use at T5 was associated with higher levels of psychosis, ODD, ADHD, and CD symptoms at T6. For substance use, higher frequency of SAM use at T5 was associated with higher levels of T6 heavy drinking, tobacco, illicit drug and marijuana use. Finally, higher frequency of SAM use was associated with greater odds of social harms, risky driving behaviors, and an AUD. There were no gender differences.

## 4. Discussion

This study is among the first to prospectively investigate how concurrent and simultaneous co-use of marijuana and alcohol relate to risks for mental health and substance use outcomes two years later among young adults, and to examine gender differences in these associations. We also investigated whether frequency of SAM (i.e., how often substances are used simultaneously rather than just experience of SAM) was differentially associated with later health outcomes. Consistent with past research, prevalence of SAM was high in our sample (41%), far exceeding the prevalence of concurrent use (13%) and marijuana only use (1%) [12,16]. Further, SAM use was associated with heavier substance use and more substance-related harm two years later compared to alcohol-only use as has been shown in past studies [6,9]. Similarly, higher frequency of SAM use was also associated with more negative mental health outcomes. However, concurrent use was also associated with problematic substance use and related harms. Gender moderation was primarily observed for the associations between co-use patterns and substance use outcomes, with female concurrent use, but not male concurrent use, associated with heavier use of alcohol and illicit drug use relative to alcohol-only use.

Compared to alcohol-only use, SAM use was associated with higher symptoms of psychosis and externalizing problems (i.e., ODD, ADHD, CD) two years later. Heavy marijuana use has been linked to psychosis symptoms across the lifespan and both alcohol and marijuana [23] have been linked to externalizing behaviors [2,33]. Externalizing problems typically precede and contribute to substance use risk [34] and frequent SAM use may in turn exacerbate externalizing behaviors in young adulthood due to the greater impairment resulting from SAM use and the risky social contexts where SAM use typically occurs [16]. Experimental clinical trials have found that SAM use results in higher THC concentrations in the blood, and an increased duration of marijuana effects resulting in greater impairment compared to either substance alone [12]. Further, Lipperman-Kreda et al. [16] found that SAM use is more likely to occur in unsupervised contexts with large numbers of underage drinkers. Youth who engage in SAM may also be more likely to come in contact with deviant peers increasing the likelihood of externalizing behaviors.

There were few differences in outcomes between concurrent use and alcohol-only use, however, concurrent use was associated with more conduct symptoms, illicit drug use, and AUD risk. Consistent with past research, these findings suggest that the risks associated with concurrent use exceed that of alcohol for highly problematic substance use and behavior problems, but may be associated with fewer negative outcomes compared to SAM use [1,6]. Further, concurrent use may pose less risk for psychosis, social harms and unsafe driving practices than SAM. Although further animal research is needed, it is possible that the co-enhancing effects of alcohol and marijuana when used simultaneously increases the likelihood of psychosis among youth engaged in SAM.

While SAM use was generally associated with heavier substance use compared to concurrent use, there were no differences in substance-related harms (except CUD) and SAM use was associated with lower levels of anxiety and depression (marginally) symptoms; no other differences in mental health symptoms were observed. Experimental clinical trials suggest that consuming alcohol and marijuana together results in pharmacological interactions that alter experiences of impairment and bioavailability of the drugs [12]. We predicted that SAM use would result in different outcomes compared to concurrent use patterns where the effects of alcohol and marijuana are independent and do not overlap. Our findings of few differences between SAM use and concurrent use patterns may reflect methodological limitations. For example, the clinical effects of SAM may be episodic and not result in cumulative risk for mental health or substance use problems over time. An ecological momentary assessment design may be needed to capture these acute SAM risks. Further, past studies have been cross-sectional and relied on co-use status as an indication of risk. These studies may overestimate risks associated with SAM use or fail to capture important heterogeneity in outcomes related to frequency of SAM use.

Indeed, we found that frequency of SAM use was associated with long term risk of more substance use, greater risk of all harm outcomes examined and psychosis and externalizing symptoms. This finding aligns with recent work identifying two types of SAM use, heavy use and lighter use, which are distinguished by frequency of alcohol and marijuana use [1]. Heavier SAM use was associated with more truancy, illicit drug use and more evenings out compared to Light SAM use and concurrent use. Terry-McElrath et al. [9] also reported that unsafe driving behaviors were higher among youth who reported regular SAM use (most or every time) than those reporting occasional SAM use. Studies largely relied on binary measures of SAM use status, but these finding suggests that the association between SAM use and health outcomes may be dose-dependent and that frequency of SAM may be a better indicator of future risk. Future research is needed to determine whether there is a threshold of risk for SAM use.

Consistent with past research, boys were more likely to be engaged in SAM [12]. Findings revealed few gender differences in risks for mental health or substance-related harms related to concurrent or SAM use. However, female concurrent use (not male concurrent use) was associated with heavier drinking and illicit drug use compared alcohol-only use. Further, male SAM use (but not female SAM use) was associated with higher frequency of all substances (except tobacco) relative to concurrent use, but lower levels of anxiety and depression. These findings indicate that the substance use profiles of male and female SAM users may differ. Specifically, concurrent use may pose more substantial substance use risks for girls, compared to boys. The observed differences may be attributable to gender disparities in exposure and vulnerability to risk factors, such as peer substance use or poor school performance, or reflect gender-specific motives for SAM use. More research is needed to understand gender differences in co-use motives and contexts to elucidate how co-use patterns differ between boys and girls.

## 5. Limitations

The longitudinal design of the current study allowed us to prospectively examine the extent to which SAM use and concurrent use of alcohol and marijuana pose risk for poor substance use and mental health outcomes two-years later. However, the findings should be interpreted in light of several limitations. First, the sample was primarily Caucasian and findings may not generalize to other ethnic groups. Second, SAM use was defined as use of alcohol and cannabis within 3 h, however, there is considerable variability in the operationalization of SAM use across studies and this may have impacted study findings. Recent research found no differences in the consequences of SAM or subjective intoxication across competing operationalizations of SAM (defined as co-use occurring within 1–240 min in increments of 1 min) giving us confidence that the study findings will generalize beyond our operationalization of SAM [17]. Third, while beyond the scope of the current study, the order of consumption (alcohol or cannabis first) may alter the consequences of SAM use and should be investigated in future studies. Fourth, data wer collected before legalization and the effects of legalization on findings are unknown. Fifth, while the study elucidates potential long-term risks of SAM use, the acute risks related to levels of intoxication for SAM users are not captured here and require observational and daily diary studies.

## 6. Conclusions

The findings suggest that any co-use (concurrent or simultaneous) may be problematic in young adulthood. Concurrent use enhances levels of substance use among users which may increase risk for addictions and frequent SAM use is associated with higher frequency and levels of all substances, not just SAM use, and are at increased risk of behavioral problems, psychosis and substance-related harms compared to alcohol-only use. The findings have several important implications for prevention, particularly with the evolving legalization of marijuana. Since marijuana is rarely consumed alone, public health messages need to inform consumers about the possible consequences of using both alcohol and marijuana and the addictive pharmacological impact of using them together. Surveillance surveys and screening tools used in primary care settings should routinely ask about SAM use to identify those at risk. The findings may also suggest that retail cannabis should not be sold in the same locations as alcohol to minimize the likelihood they are purchased and consumed together. To better target prevention and intervention programs to SAM use future research is needed to increase our understanding of why young people choose to consume alcohol and marijuana simultaneously.

## Figures and Tables

**Table 1 ijerph-18-03652-t001:** Means of Substance Use Variables by Time 5 Use Groups.

		Alcohol-Only (*n* = 189; 42%)	Concurrent (*n* = 60; 13%)	Simultaneous (*n* = 182; 41%)	
	Range	Mean (SD) or *n* (%)	Mean (SD) or *n* (%)	Mean (SD) or (%)	*F* or χ^2^
Gender					
Boys		74 (39%) ^a^	21 (35%) ^a^	108 (59%) ^b^	19.26 ***
Girls		115 (61%)	39 (65%)	74 (41%)	
Substance Use History					
Age of Onset Alcohol	10–23	15.05 (2.48) ^a^	14.00 (2.02) ^b^	14.08 (1.87) ^b^	10.91 ***
Age of Onset Marijuana	6–26	16.22 (2.67) ^a^	15.87 (2.73) ^ab^	15.74 (2.56) ^b^	4.38 *
T1 Freq of HED	1–5	1.43 (0.83) ^a^	1.67 (1.10) ^ab^	1.83 (1.14) ^b^	7.34 ***
T1 Freq of Marijuana	0–4	1.47 (0.91) ^a^	1.70 (1.06) ^ab^	1.98 (1.37) ^b^	9.09 ***
Mental Health T6					
Anxiety	0–12	5.40 (2.93) ^a^	6.37 (2.93) ^b^	5.19 (2.38) ^a^	3.85 *
Depression	0–12	2.68 (2.60)	3.44 (2.81)	2.75 (2.34)	1.99
Psychosis	0–10	1.70 (2.15) ^a^	2.07 (2.21) ^ab^	2.58 (2.35) ^b^	6.56 **
ADHD	0–11	3.39 (2.28) ^a^	4.00 (2.65) ^ab^	3.99 (2.29) ^b^	3.21 *
ODD	0–11	2.67 (2.03) ^a^	3.37 (2.43) ^b^	3.24 (2.11) ^b^	4.06 *
CD	0–7	0.80 (1.00) ^a^	1.13 (0.97) ^a^	1.25 (1.06) ^b^	8.33 ***
Substance Use T6					
Freq of HED	0–4	1.03 (1.08) ^a^	1.30 (1.19) ^b^	1.94 (1.14) ^b^	28.11 ***
Cigarettes Past week	0–4	0.36 (1.07)	0.55 (1.29)	0.97 (1.51)	9.53 ***
Freq of Marijuana	0–4	0.20 (0.55)	1.09 (1.23)	2.13 (1.43)	131.55 ***
Freq of Illicit Drugs	0–10	0.21 (0.63) ^a^	0.44 (0.63) ^b^	0.63 (0.79) ^b^	15.14 ***
Substance-related Harms T6					
Social harms		37 (21%) ^a^	17 (32%) ^ab^	66 (41%) ^b^	16.04 ***
Alcohol Use Disorder		45 (25%) ^a^	23 (43%) ^b^	91 (57%) ^c^	33.93 ***
Marijuana Use Disorder		2 (1%) ^a^	5 (10%) ^b^	48 (30%) ^c^	56.90 ***
Driving/riding drunk		35 (20%) ^a^	17 (32%) ^ab^	58 (36%) ^b^	11.39 **
Driving/riding high		9 (5%) ^a^	13 (24%) ^b^	69 (43%) ^c^	67.92 ***

Note: Shared superscripts (a,b,c) indicate no significant difference between group means. * *p* < 0.05, ** *p* < 0.01, *** *p* < 0.001.

**Table 2 ijerph-18-03652-t002:** Associations between substance use patterns at T5 and mental health outcomes, substance use behaviors and related harms at T6.

Dependent Variables	Model 1				Model 2		
	Alcohol (Ref)	Concurrent	Simultaneous	R^2^	Concurrent (Ref)	Simultaneous	R^2^
Mental Health Symptoms		Est (SE)	Est (SE)			Est (SE)	
Anxiety	-	0.10 (0.06)	−0.01 (0.05)	0.158	-	−0.15 (0.07) *	0.219
Depression	-	0.10 (0.06)	0.03 (0.05)	0.142	-	−0.12 (0.08) +	0.210
Psychosis	-	0.05 (0.05)	0.17 (0.05) ***	0.120	-	0.08 (0.07)	0.320
ODD	-	0.10 (0.05)	0.11 (0.05) *	0.153	-	−0.02 (0.07)	0.214
ADHD	-	0.08 (0.05)	0.11 (0.05)	0.127	-	−0.04 (0.08)	0.145
CD	-	0.11 (0.05) *	0.19 (0.06) ***	0.042	-	0.03 (0.07)	0.015
Substance Use		Est (SE)	Est (SE)			Est (SE)	
HED	-	0.07 (0.05)	0.32 (0.05) ***	0.184	-	0.20 (0.07) **	0.120
Tobacco use	-	0.05 (0.05)	0.16 (0.05) **	0.146	-	0.09 (0.07)	0.104
Illicit drug use	-	0.10 (0.05) *	0.26 (0.06) ***	0.096	-	0.11 (0.06) +	0.028
Marijuana	-	-	-		-	0.29 (0.06) ***	0.137
Substance-related Harm		Odds Ratio (95% CI)	Odds Ratio (95% CI)			Odds Ratio (95% CI)	
Social harms	-	1.53 (0.77–3.0)	2.37 (1.42–3.94) ***	0.071	-	1.55 (0.79–3.04)	0.028
AUD symptoms	-	2.05 (1.07–3.93) *	2.96 (1.81–4.83) ***	0.049	-	1.52 (0.78–2.93)	0.022
Driving/riding drunk	-	1.75 (0.88–3.49)	1.97 (1.18–3.29) **	0.117	-	1.11 (0.56–2.18)	0.134
Driving/riding high	-	-	-		-	1.95 (0.94–4.00)	0.049

Note: Models control for sex, age at baseline, parent SES, baseline measures of dependent outcomes where available. Baseline levels of psychosis, and substance-related harms were unavailable. Baseline levels of alcohol and marijuana frequency were controlled for models assessing group differences substance-related harms. Marijuana use disorder could not be included as a harm in the path model 2 because too few concurrent users met the criteria. * *p* < 0.05, ** *p* < 0.01, *** *p* < 0.001, “+” *p* < 0.10.

**Table 3 ijerph-18-03652-t003:** Associations between SAM frequency at T5 and mental health outcomes, substance use behaviors and related harms at T6.

Dependent Variables	SAM Frequency Est (SE)	R2
Mental health symptoms		
Anxiety	−0.01 (0.05)	0.190
Depression	0.06 (0.05)	0.231
Psychosis	0.18 (0.05) ***	0.292
ODD	0.11 (0.05) *	0.218
ADHD	0.11 (0.05) *	0.151
CD	0.16 (0.06) **	0.045
Substance use		
HED	0.32 (0.05) ***	0.195
Tobacco use	0.20 (0.06) **	0.140
Illicit drug use	0.28 (0.06) ***	0.152
Marijuana	0.67 (0.04) ***	0.513
Harms	OR (95% CI)	
Social harms	1.28 (1.05–1.56) *	0.049
AUD symptoms	1.41 (1.15–1.73) **	0.053
Driving/riding drunk	1.38 (1.12–1.70) **	0.247
Driving/riding high	2.04 (1.62–2.58) ***	0.090

Note: * *p* < 0.05, ** *p* < 0.01, *** *p* < 0.001; Model control for parent SES, age, sex, baseline levels of the dependent variable, and baseline levels of alcohol frequency and cannabis frequency.

## Data Availability

The data presented in this study are available on request from the corresponding author. The data are not publicly available as they are archival and ethics approval was not obtained at the time when the data were collected or the informed consent secured that would allow for posting the data in a publicly accessible repository.

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
