# Peer review of "Co-Use of Alcohol and Cannabis: Longitudinal Associations with Mental Health Outcomes in Young Adulthood"

_ijerph, 2021, doi:10.3390/ijerph18073652_

Round 1
Reviewer 1 Report
We think this is an interesting proposal, both from the point of view of scientific knowledge and from the point of view of social impact.
We propose that the authors should review the following comments.
Firstly, we propose some small modifications in the format of the text:
- The identification of section 3.3 (246) is repeated.
- The text 353-364 is repeated in the conclusions (365-377).
- We propose to clarify the structure of the manuscript. For example, in fragment 84-102, the tables showing the results of the three research pathways could be indicated.
- We propose to remove the reference to the legalization of cannabis (first sentence of the abstract) because it may confuse the reader about the mains of this paper. Data from this investigation predate the legalization.
- We propose to modify the term “effects”, as it can be confusing and wrongly assume that causal relationships have been identified. It should be made clear that the authors are evaluating association, not causality.
- The resolution and visualisation of the Tables should be revised.
Secondly, there are general technical comments:
- The authors examined the association between concurrent use (or alcohol-only) and mental health problems. It is not clear whether the authors have controlled the quantity/frequency of consumption. If it left unchecked, the mental health problems could be due to a higher consumption, not strictly to simultaneous consumption. If so, we propose two alternatives: a) to adapt it, or b) to alert it as a limitation (text can be taken as a reference: 327-331).
- Authors should report the sampling error of the survey.
- The methodology chapter indicates the sample size at T6 is larger than at T5 (n = 464 at T5 and n = 478 at T6). We understand that some young people may not participate in one time (e.g., T5) and re-participate in the next time (T6). Is it correct? We understood that the sample is made up of the same people. Is it correct? So, it should be noted that the survey had a panel format.
- The outcomes (T6) are evaluated two years later (T5). Do the authors consider that is enough time to obtain their conclusions? Can the authors justify why haven’t they taken a wider longitudinal frame (e.g., T1)?
- The T5 data were collected in 2011 and the T6 data were collected from 2013. Although 8-10 years have passed, and the legalization of cannabis in Canada has been later (2018), can the authors justify the validity of the research?
- In the results, the authors should report the effect size (not just the p-value) and post-hoc contrasts (in ANOVA models).
- What are the thresholds for psychotic symptoms? (136-137).
Table 1
- Some results in the Table 1 do not match those in paragraph (196-199). For example: concurrent use: 13% in text and 14% in table.
- What do the notes * in the results in Table 1 mean? (there is no interpretative note).
- Last column: we read “F or” (does this mean “F or K Kruskal Wallis”, or simply “F”?)
Table 2
- It should be noted that in many cases there are no differences, especially between alcohol and concurrent users.
- SAM use is associated with lower levels of anxiety (men only). Can the authors identify some reason?
- On the other hand, model 2 in Table 2 does not indicate a relationship with depression levels. Which of the two statements (text or table) is correct?
Conclusions
- 372-374. The authors conclude that public health messages should report more explicitly on the effects of alcohol and cannabis use. However, another study published in IJERPH in 2020 indicates that young people who perceive themselves to be well-informed have the highest prevalence of alcohol and cannabis consumption (they describe it as an information paradox). The authors defend other complementary measures in addition to the information, for example: social intervention in schools should go beyond than simply providing briefings about drugs, and they defend the importance of involving families in educational prevention, combined with responsible supervision. Do the authors agree with this extension?
Author Response
Reviewer 1:
We think this is an interesting proposal, both from the point of view of scientific knowledge and from the point of view of social impact.
We propose that the authors should review the following comments.
Firstly, we propose some small modifications in the format of the text:
- The identification of section 3.3 (246) is repeated.
Response: This error has been fixed.
- The text 353-364 is repeated in the conclusions (365-377).
Response: Line 361-372 deleted.
- We propose to clarify the structure of the manuscript. For example, in fragment 84-102, the tables showing the results of the three research pathways could be indicated.
Response: We are not entirely clear what the reviewer is asking of us here. However, we feel that any reference to the results tables in the introduction is unusual and not appropriate.
- We propose to remove the reference to the legalization of cannabis (first sentence of the abstract) because it may confuse the reader about the mains of this paper. Data from this investigation predate the legalization.
Response: See Abstract, Line 9
- We propose to modify the term “effects”, as it can be confusing and wrongly assume that causal relationships have been identified. It should be made clear that the authors are evaluating association, not causality.
Response: We have replaced the term “effects” through the paper when about the current study.
- The resolution and visualisation of the Tables should be revised.
Response: The tables have been saved as optimized pdf. Zipped and regular version.
  Secondly, there are general technical comments:
- The authors examined the association between concurrent use (or alcohol-only) and mental health problems. It is not clear whether the authors have controlled the quantity/frequency of consumption. If it left unchecked, the mental health problems could be due to a higher consumption, not strictly to simultaneous consumption. If so, we propose two alternatives: a) to adapt it, or b) to alert it as a limitation (text can be taken as a reference: 327-331).
Response: Frequency of alcohol and cannabis were controlled for. This has been clarified in Line 198. It was also previously detailed in the notes under the tables.
- Authors should report the sampling error of the survey.
Response: Is the reviewer is asking for the response rate? It is not statistically possible to provide a sampling error without knowing the true values of the population the sample was drawn from. Response for T5 and T6 rate has been provided on Line 189 and 190.
- The methodology chapter indicates the sample size at T6 is larger than at T5 (n = 464 at T5 and n = 478 at T6). We understand that some young people may not participate in one time (e.g., T5) and re-participate in the next time (T6). Is it correct? We understood that the sample is made up of the same people. Is it correct? So, it should be noted that the survey had a panel format.
Response: Yes, this is correct. This is a cohort-sequential longitudinal design, thus participants occasionally missed timepoints but were invited to participate at each time. In Line 116 we have included that this is a panel study.
- The outcomes (T6) are evaluated two years later (T5). Do the authors consider that is enough time to obtain their conclusions? Can the authors justify why haven’t they taken a wider longitudinal frame (e.g., T1)?
Response: Yes. The SAM use measure was only introduced in T3. We were also only interested in these associations in young adulthood.
- The T5 data were collected in 2011 and the T6 data were collected from 2013. Although 8-10 years have passed, and the legalization of cannabis in Canada has been later (2018), can the authors justify the validity of the research?
Response: Legalization has heighted public attention to cannabis use and its associated risks, but the prevalence of use among young people has been consistently higher than at any other time in the lifespan for decades. Comparing our data to the recent Canadian Cannabis survey data shows that prevalence rates in our study from 2011 (55%) are comparable to current national use rates of use among young adults 20-24 years old in 2020 (52%), suggesting that the profile of cannabis has remained high and consistent in this population despite legalization. (https://www.canada.ca/en/health-canada/services/drugs-medication/cannabis/research-data/canadian-cannabis-survey-2020-summary.html#a2 ).
- In the results, the authors should report the effect size (not just the p-value) and post-hoc contrasts (in ANOVA models).
Response: Effect sizes have been added. ANOVA models were run to compare means in our dependent variables across substance use groups. The F value indicates the overall significance of the AVONA and the significance of the post-hoc contrasts are denoted by shared subscripts. We believe the additional of the coefficients of the post-hoc comparisons is unnecessarily and cumbersome given the volume of contrasts.
- What are the thresholds for psychotic symptoms? (136-137).
Response: Details about how items for psychotic symptoms are coded have been added to line 148-149. Clinical cut-offs are not available for SCL-90-R
Table 1
- Some results in the Table 1 do not match those in paragraph (196-199). For example: concurrent use: 13% in text and 14% in table.
Response: This has been corrected in Table 1.
- What do the notes * in the results in Table 1 mean? (there is no interpretative note).
Response: An interpretive note has been added to Table 1.
- Last column: we read “F or” (does this mean “F or K Kruskal Wallis”, or simply “F”?)
Response: This was intended to indicate F values (for ANOVA) or X2 values for the binary DVs. This has been corrected in Table 1.
Table 2
- It should be noted that in many cases there are no differences, especially between alcohol and concurrent users.
Response: This is stated in line 357
- SAM use is associated with lower levels of anxiety (men only). Can the authors identify some reason?
Response: Yes, this finding was unexpected. These gender differences may be driven by differences in SAM use motives between men and woman. For example, males may primarily engage in SAM use for pleasure, enjoyment, or enhance their high, whereas women may be more apt to use SAM to cope with negative affect. We have include this hypothesis in the discussion. See line 398-403.
- On the other hand, model 2 in Table 2 does not indicate a relationship with depression levels. Which of the two statements (text or table) is correct?
Response: We have clarified this on lines 240 to 241 and added a “+”to Table 2 to indicate that the association between SAM use and depression was marginally significant.
Conclusions
- 372-374. The authors conclude that public health messages should report more explicitly on the effects of alcohol and cannabis use. However, another study published in IJERPH in 2020 indicates that young people who perceive themselves to be well-informed have the highest prevalence of alcohol and cannabis consumption (they describe it as an information paradox). The authors defend other complementary measures in addition to the information, for example: social intervention in schools should go beyond than simply providing briefings about drugs, and they defend the importance of involving families in educational prevention, combined with responsible supervision. Do the authors agree with this extension?
Response: The information paradox is an interesting phenomenon. I agree that providing information in and of itself may not be effect at minimizing use or risk of harm. As a solo strategy, decades of research have shown that education and knowledge about harm associated with substance use is not an effective deterrent for young people. However, from a public health perspective, providing accurate public health messages at allow those who use cannabis to make informed choices regarding their substance use (whether they actually do or not) is a consumer right. These findings may also inform changes in cannabis policies, such as not selling alcohol and cannabis in the same retail location. Line 451- 455.
Reviewer 2 Report
Thank you for giving me the opportunity to read this valuable paper, the concurrent/simultaneous use of alcohol and cannabis is a topic of growing concern and an area where more research is needed especially with the increase in legalisation of cannabis internationally. Overall I feel this publication will be of interest to the audience and add value to the research in this area. I don't have any major concerns but there are some minor points I would like to see addressed:
- Abstract, Results, Lines 14 to 15 - these numbers add up to over 100% and are not the same numbers in the results section (Line 196 and 197). Please correct
- Introduction, Line 42 -43 - I could recommend editing the sentence "and would inform the development of targeted educational messages and policy reform." as it implies that policy reform will be needed. Whilst this may be true, there isn't enough evidence to support this statement at the moment - so I would recommend either removing the word reform or updating to "and any required policy reforms"
- Introduction, Line 47 - the term "dependence" is used here however, this is the only time it is used and the rest of the publication uses "use disorder". I would recommend updating it to substance use disorder to maintain consistency.
- Line 51 - increased instead of increase?
- Line 63 to 65 - associated used twice in the same sentence. Perhaps change the first use to linked?
- Use of "sex" rather than "gender" - was the wording in the survey "what sex were you assigned at birth". Or did the question only ask if the person was male/female? In which case - was any thought given to people who identify as a different gender?
- Materials & Methods Line 114 - Mage seems an old way to say mean age, is there a more clear way to shorten this? Perhaps age (M)?
- Line 130 - first use of the ODD, please define
- Line 142 - what is meant by "club drugs"? Any of the other drugs listed could be considered club drugs particularly amphetamine. Please provide more information or examples of what drugs fell into this category
- Table 1 - Footnote with explanatory notes on symbols etc seems to be missing.
- Line 215 - first use of CD, please define.
- Line 248 - first use of AUD, please define.
- Line 251-252 - add in acroynm CUD. although in Line 305 you refer to it as marijuana use disorder - please use a consistent term throughout.
- Discussion was thorough in my opinion, however, I would like to see more discussion around the intricacies of simultaneous use (perhaps suited to limitations section?). This paper defined simultaneous use as "within 3 hours" - what was the rationale for that? Are there any studies around whether timing (e.g. taking it in the same hour vs a couple of hours later) has an impact? Or is this an area where more investigation is needed?
- Also what about the order in which the drugs were taken - this might have been out of the scope of this current study, however, I would be interested to see this commented on in the discussion (even if it is a comment that says further investigation is needed).
Author Response
Reviewer 2:
Thank you for giving me the opportunity to read this valuable paper, the concurrent/simultaneous use of alcohol and cannabis is a topic of growing concern and an area where more research is needed especially with the increase in legalisation of cannabis internationally. Overall I feel this publication will be of interest to the audience and add value to the research in this area. I don't have any major concerns but there are some minor points I would like to see addressed:
- Abstract, Results, Lines 14 to 15 - these numbers add up to over 100% and are not the same numbers in the results section (Line 196 and 197). Please correct
Response: Thank you for noting this typo. We have corrected it in the abstract (line 14) and in the results (line 196-197).
- Introduction, Line 42 -43 - I could recommend editing the sentence "and would inform the development of targeted educational messages and policy reform." as it implies that policy reform will be needed. Whilst this may be true, there isn't enough evidence to support this statement at the moment - so I would recommend either removing the word reform or updating to "and any required policy reforms"
Response: We have removed the word reform as recommended from Line 44.
- Introduction, Line 47 - the term "dependence" is used here however, this is the only time it is used and the rest of the publication uses "use disorder". I would recommend updating it to substance use disorder to maintain consistency. 
Response: We have replaced “dependence” with “substance use disorders” in line 55.
- Line 51 - increased instead of increase?
Response: We changed this to “increases in” – see line 59.
- Line 63 to 65 - associated used twice in the same sentence. Perhaps change the first use to linked?
Response: We replaced the second “associated risks” with “negative consequences” – line 73.
Use of "sex" rather than "gender" - was the wording in the survey "what sex were you assigned at birth". Or did the question only ask if the person was male/female? In which case - was any thought given to people who identify as a different gender? 
Response: The survey asks participants to report whether they are male/female/transgender, therefore all reference to sex has been changed to gender. One participants identified as transgendered and was excluded from sex-based analysis.
- Materials & Methods Line 114 - Mage seems an old way to say mean age, is there a more clear way to shorten this? Perhaps age (M)?
Response: Mage has been replaced with M = 15.5 years old. Line 123.
- Line 130 - first use of the ODD, please define
Response: Corrected – see line 147.
- Line 142 - what is meant by "club drugs"? Any of the other drugs listed could be considered club drugs particularly amphetamine.  Please provide more information or examples of what drugs fell into this category
Response: Club drugs has been defined on like 162.
- Table 1 - Footnote with explanatory notes on symbols etc seems to be missing. 
Response: Footnote for Table 1 is added.
- Line 215 - first use of CD, please define. 
Response: CD is now defined on line 149.
- Line 248 - first use of AUD, please define. 
- Response: AUD is now defined on line 191.
- Line 251-252 - add in acroynm CUD. although in Line 305 you refer to it as marijuana use disorder - please use a consistent term throughout. 
Response: CUD is defined on line 191. We now consistently refer to CUD or cannabis use disorder throughout the manuscript.
- Discussion was thorough in my opinion, however,  I would like to see more discussion around the intricacies of simultaneous use (perhaps suited to limitations section?). This paper defined simultaneous use as "within 3 hours" - what was the rationale for that? Are there any studies around whether timing (e.g. taking it in the same hour vs a couple of hours later) has an impact? Or is this an area where more investigation is needed?
Response: There is considerable variability in the operationalization of SAM use across studies, but a recent study testing differences in consequences and subjective intoxication found no differences across different operationalizations. We have included this in the limitations section Line 388 - 394
- Also what about the order in which the drugs were taken - this might have been out of the scope of this current study, however, I would be interested to see this commented on in the discussion (even if it is a comment that says further investigation is needed). 
Response: This has been added to the limitations section on Lines 394 – 396.